# PLAN-BASED PROMPTING IMPROVES LITERATURE REVIEW GENERATION

## ABSTRACT

We explore the zero-shot abilities of recent large language models (LLMs) for the task of writing the literature review of a scientific research paper conditioned on its abstract and the content of related papers. We propose and examine a novel strategy for literature review generation with an LLM in which we first generate a plan for the review, and then use it to generate the actual text. While modern LLMs can easily be trained or prompted to condition on all abstracts of papers to be cited to generate a literature review without such intermediate plans, our empirical study shows that these intermediate plans improve the quality of generated literature reviews over vanilla zero-shot generation. Furthermore, we also create a new test corpus consisting of recent arXiv papers (with full content) posted after both open-sourced and closed-sourced LLMs that were used in our study were released. This allows us to ensure that our zero-shot experiments do not suffer from test set contamination.

## 1 INTRODUCTION

We are interested in understanding the behaviour and potential of large language models (LLMs) for assisting with the generation of literature reviews of scientific papers. The task of writing a literature review — finding, citing and contextualizing relevant prior work is an important part of the scientific process. This challenge is exacerbated in machine learning due to the current extremely rapid pace of progress, with relevant papers appearing every day on the arXiv. However, naively applying LLMs can hallucinate content and even cite imaginary papers that do not exist. We address this key problem by operating in the setting were LLMs are prompted to only use information from a specific set of actual papers to be cited, and we provide the abstracts of those papers as conditioning to our models. Furthermore, we explore a decomposition of the task where we provide a writing plan consisting of which papers to cite at which points in the literature review. Our experiments indicate that with this strategy it is possible to unlock even higher levels of quality.

Prior work has shown that it is possible to use LLMs to automatically generate a high-quality abstract for a scientific paper, conditioned on the rest of the paper (Pilault et al., 2020). Many other subsequent methods such as the well-known BART (Lewis et al., 2020) and PEGASUS (Zhang et al., 2020) models have also demonstrated strong results for the tasks of news (Grusky et al., 2018a) and patent summarization (Sharma et al., 2019a). In contrast to these and other single document abstractive summarization tasks, when writing literature reviews the potential input context is much longer. Even if the papers to be cited have already been identified, summarizing and/or contextualizing the content of all papers to be cited represents a challenging variation of the classical "multi-document summarization" problem. The scale of this problem setting poses challenges for the application of certain types of currently popular few-shot prompting (Brown et al., 2020) techniques, few-shot Chain-of-thought prompting (Wei et al., 2022) and in-context prompting techniques aka "Prompt Engineering" (e.g. Li & Liang, 2021; Qin & Eisner, 2021). However our work here proposes a viable and empirically effective solution. In our approach we either task the LLM to first produce a writing plan which it should then follow to generate the output, or we allow the user to provide the plan as additional prompting — or some combination of the two in the case of using our approach within an interactive, iterative writing assistance system. We consider our proposed approach here as a first step towards building next-generation intelligent conversational agents which help researchers write the related work section of scientific papers or potentially whole survey papers having interactive editing capabilities (Dwivedi-Yu et al., 2022).

Our main contributions are:

- We propose and examine the use of a writing-plan-based intermediate task to improve the quality of the final generated text. Our prompt engineering techniques are reminiscent of the traditional modular pipelines of Natural Language Generation (consisting of content planning, sentence planning and surface realization), and provide clear improvements to the quality of generated text compared to vanilla text generation.

- We create a new corpus consisting of recent papers (with full content) published on the arXiv after both open-sourced and closed-sourced LLMs were released. This allows us to properly probe the zero-shot setting without the risk of test set contamination. We intend to release this dataset to the academic community.

- We provide experimentation comparing prompt engineering techniques with task-specific fine-tuning and benchmark the performance of open-sourced LLMs in both settings. Our experiments with closed-sourced and open-sourced LLMs reveal a significant gap, with closed-sourced LLMs (GPT-4) still outperforming the latest open-sourced LLMs (Llama 2-Chat) in zero-shot setting, both in terms of automatic metrics as well as LLM based evaluation setup (Zheng et al., 2023).

## 2 Our Approach & New Datasets

### 2.1 Our Plan Based Generation Approach & Model Variants

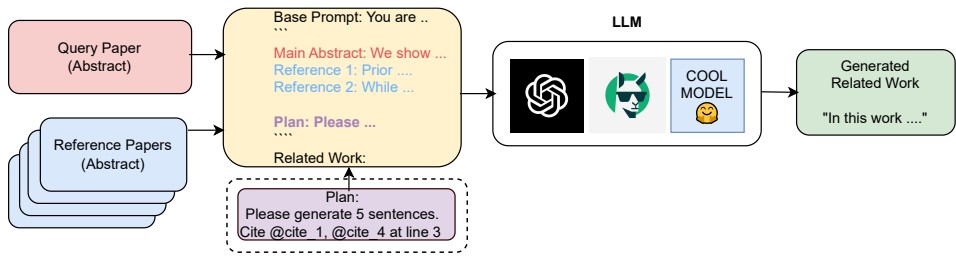

Figure 1: Pipeline of our task where the model needs to generate the related work of the query paper conditioned on reference papers. Our method employs an optional plan – shown by the dotted purple box, which is either generated by the model or appended to the prompt to generate the output.

We work under the framework shown in Figure 1. The task is to use the abstract of a query paper and the abstracts of a set of reference papers to generate the related works section of the query paper. Our approach relies on prompting an LLM to achieve this task.

There are several dimensions under which such a framework (and scientific writing in general) can be evaluated. While it is possible that the discourse quality of recent LLMs will yield well-written passages, their factual accuracy might be weaker as LLMs are known to "hallucinate" .

In this work, we propose to decompose the writing task to increase both passage quality and factual accuracy. We propose different methods for generating a *writing plan*, a line-by-line description including citations of the passage to write. These writing plans also provide a level of control over the output passages to authors (users). This is likely to be essential in practice to meet author preferences and possible publication constraints. In the following, we describe our methods concretely.

**Plan-based** In plan-based generation, the model is prompted with a plan to produce X sentences in Y words and cite references on respective lines derived from the GT related work. 4). An example of the format of these plans is provided below:

```
Please generate {num_sentences} sentences in {num_words} words.  Cite {cite_x} at line
{line_x}.  Cite {cite_y} at line {line_y}.
```

**Learned plan** We envision use case scenarios where a plan is optionally provided by the author. In case it is not, we also propose to learn the plan generation. The model is prompted to first generate a

plan of sentences and citations which it would then condition upon to generate the final related work text. When used as an interactive tool we envision the user might start with a suggested plan, see the corresponding generated full literature review text, and then iteratively edit the plan and regenerate the result. See our Appendix for the different prompts (eg. Figure 5)

We also experiment with 2 other strategies in which researchers could prompt the model:

**Per cite** We first use a two-stage strategy to generate content relevant to each cited paper. In the first stage, the model is prompted to generate related works in 1-2 lines for each individual reference citation. All the outputs for different citations are combined together to form the generated related work. In the second stage, the LLM summarizes and paraphrases the output of the first stage.

**Sentence by sentence** Based on the Ground truth (GT) related work and the citation on each line, we prompt the model to generate one sentence conditioned on the abstract, the reference cited in that line, and the generated draft so far. In the absence of a citation or at the start, the model is prompted only with abstract and draft of the generated work till now.

## 2.2 NEW DATASETS

| Dataset | Task |
|---|---|
| BigSurvey-MDS (Liu et al., 2023) | Survey Introduction |
| HiCaD (Zhu et al., 2023) | Survey Catalogue |
| SciXGen (Chen et al., 2021a) | Context aware text generation |
| CORWA (Li et al., 2022) | Citation Span Generation |
| TLDR (Cachola et al., 2020) | TLDR generation |
| Multi-XScience Lu et al. (2020) | Related Work Generation |

Table 1: Different tasks for academic literature

While there are datasets available for different tasks in academic literature (see Table 1), we use the Multi-XScience dataset (Lu et al., 2020) for our experiments. Recent work (Chen et al., 2021b; Funkquist et al., 2022) also focuses on related work generation and provides a similar dataset. As part of this work, we release two corpora: 1. We extend the Multi-XScience corpus to include full-text of research papers and 2. We create a new test corpus (MXS-2308) consisting of recent (August 2023) arXiv papers (with full content).

**Multi-XScience full text** We create these datasets based on the latest release (2023-09-12) of the S2ORC corpus[1] (Lo et al., 2020) available at the Semantic Scholar Open Data Platform (Kinney et al., 2023). The S2 Platform provides access to multiple datasets including papers metadata, authors, S2AG (Semantic Scholar Academic Graph), paper embeddings, etc. While the 'Papers' dataset consists of 200M+ metadata records, S2ORC consists of 11+M full-text publicly available records with annotations chunked into 30 files (~215G compressed json) where research documents are linked with arXiv and Microsoft Academic Graph (MAG) (Sinha et al., 2015) IDs, when available. This corpus provides full text of the research papers (parsed using a complex pipeline consisting of multiple LaTeX and PDF parsers such as GROBID (Lopez, 2023) and in-house parsers.[2]). The full text is also aligned with annotation spans (character level on the full text), which identify sections, paragraphs, and other useful information. It also includes spans for citation mentions, and the matching semantic corpus-based ID for bibliographical entries making it easier to align with references compared to other academic datasets such as LoRaLay (Nguyen et al., 2023), UnarXive (Saier & Färber, 2020; Saier et al., 2023), etc. or relying on citation graphs like OpenAlex (Priem et al., 2022), next-generation PDF parsers (Blecher et al., 2023) or other HTML webpages.[3] For the Multi-XScience, we obtain the full text of papers for 85% of records from the S2ORC data.

**MXS-2308** Llama 2 was publicly released on 18th July 2023 and GPT-4 on 14 March 2023. Both provide limited information about their training corpus where academic texts in the Multi-XScience may or may not have been part of their training data. To avoid overlap with the training data of

---

[1]Dataset available at http://api.semanticscholar.org/datasets/v1/

[2]https://github.com/allenai/papermage

[3]https://ar5iv.labs.arxiv.org/ and https://www.arxiv-vanity.com/

these LLMs, we process a new dataset using papers posted after their release date. To do so, we first filter the papers published in August 2023 from S2ORC that contain an arXiv ID resulting in ~15k papers. S2ORC does not provide the publication date of the papers directly, so we use regex '2308' on the arXiv ID to extract papers posted in 08'23. We then use section annotations to get the section names and match using synonyms ('Related Work, Literature Review, Background') to extract section spans. We take the rest of the text as conditioning context except the related work section which results in ~4.7k documents. Using the citation annotations, we extract the full text of cited papers from the S2ORC corpus again using corpus ID. Similar to Multi-XScience, we use paragraph annotations to create a dataset for the latest papers (~6.2k rows). We create a subset of 1,000 examples (MXS-2308) where we have the content of all the cited papers. We will release all the processing code and dataset versions upon publication.

## 3 RELATED WORK

Multi-document summarization (MDS) involves grounding from given multiple context references to generate responses. While there exist multiple datasets in different domains for single document summarization, e.g. Newsroom (Grusky et al., 2018b), CNN/DailyMail (Hermann et al., 2015), XSum (Narayan et al., 2018) BigPatent (Sharma et al., 2019b), there are datasets and benchmarks for MDS such as Multi-News (Fabbri et al., 2019) and WikiSum (Liu et al., 2018).

The concept of literature review generation using large language models (LLMs) is built upon the foundation laid by the Multi-XScience dataset proposed by Lu et al. (2020). This dataset paves the way for the challenging task of multi-document summarization, specifically focusing on generating the related-work section of a scientific paper. As underlined by Lu et al. (2020), this approach favors abstractive models, which are well suited for the task. However, unlike the approach suggested by Lu et al. (2020), our work introduces the use of intermediate plans to improve the quality of generated literature reviews. The empirical evidence presented in our study shows that our novel strategy outperforms the vanilla zero-shot generation previously championed by the Multi-XScience dataset (Lu et al., 2020). (Note: This paragraph was entirely generated by GPT-4 following plan-based generation.[4])

Closely related to our work, Gao et al. (2023) generates answers for questions based on the citations from Wikipedia. Also related to our work, Pilault et al. (2020) examined LLM based abstractive summarization of scientific papers in the Arxiv dataset of Cohan et al. (2018); however, their work was limited to creating the abstract of a single document. Perhaps the most similar prior prompting-based approach to our work is known as 0-shot chain-of-thought prompting (Kojima et al., 2022; Zhou et al., 2022) where a model is prompted with 'Let's think step-by-step' (and similar prompts).

Traditional methods for Natural Language Generation have typically employed a rule-based modular pipeline approach comprising of multiple stages of generation with intermediary steps of content planning (selecting content from input while also determining the structure of the output), sentence planning (planning the structure of sentences) and surface realization (surfacing the text in sentence) (Reiter & Dale, 1997; Stent et al., 2004; Walker et al., 2007). Our proposed plan based prompting technique draws a parallel between the modern methods of end-to-end neural models for joint data-to-text generation with micro or content planning (Gehrmann et al., 2018; Puduppully et al., 2019; Puduppully & Lapata, 2021).

Additionally, Galactica, a large language model, has been developed to store, combine, and reason about scientific knowledge (Taylor et al., 2022). It outperforms existing models on various scientific tasks and sets new state-of-the-art results on downstream tasks. These findings highlight the potential of language models as a new interface for scientific research. However, the Galactica model was not developed to specifically address the problem of literature review assistance and it was not instruction fine-tuned to follow writing plans, and as such it suffered from the effect of hallucinating non-existent citations and results associated with imaginary prior work.[5] However, our study focuses on exploring the zero-shot abilities of LLMs for literature review generation and proposes a novel strategy that includes generating an intermediate plan before generating the actual text. Our empir-

---

[4]We use the plan: Please generate 5 sentences in 60 words. Cite @cite_1 at line 1, 3 and 5. We postprocess to replace delexicalized tokens with latex commands. Outputs from other models are compared later in Appendix.

[5]This sentence was inserted by the authors.

ical study shows that these intermediate plans improve the quality of generated literature reviews compared to vanilla zero-shot generation. Furthermore, we ensure the validity of our experiments by using a new test corpus consisting of recent arXiv papers to avoid test set contamination. (Note: This paragraph was generated by GPT-3.5 with the 4th sentence added by the authors).

## 4 EXPERIMENTS

### 4.1 IMPLEMENTATION

We use PyTorch (Paszke et al., 2017) for our experiments.[6] Specifically, we use HuggingFace Transformers library with bitsandbytes integration which allows for mixed-precision, quantized and LoRA training (Micikevicius et al., 2017; Dettmers, 2016; Dettmers et al., 2022; Dettmers & Zettlemoyer, 2022; Hu et al., 2021; Dettmers et al., 2023). We use the Adam optimizer (Kingma & Ba, 2014) for fine-tuning and calculate ROUGE scores (Lin, 2004) using the Huggingface (Wolf et al., 2019) evaluate library[7]. To split sentences, we use 'en_core_web_sm' model from SpaCy[8] . Additionally, we use Anyscale endpoints[9] to generate 0-shot Llama 2 results and OpenAI API[10] to generate results for GPT-3.5-turbo and GPT-4. For reporting results with longer context Llama 2 using RoPE scaling (Su et al., 2021), we use HuggingFace Text Generation Inference.[11]

### 4.2 BASELINES

**Extractive baselines** As in Lu et al. (2020), we report the performance of LexRank (Erkan & Radev, 2004) and TextRank (Mihalcea & Tarau, 2004). We also create a simple one-line extractive baseline which extracts the first line of the abstract and combines all the citations together to form the output.

**Abstractive finetuned baselines** We use the model outputs of Hiersum (Liu & Lapata, 2019) and Pointer-Generator (See et al., 2017) from Lu et al. (2020) for abstractive finetuned baselines. We also reproduce the finetuned PRIMER (Xiao et al., 2021) model (considered to be SOTA).

**Abstractive zero-shot baselines** We further consider the zero-shot single-document abstractive summarizers FlanT5 (Chung et al., 2022) and LongT5 (Guo et al., 2022) based on the T5 architecture (Raffel et al., 2020). Since Galactica (Taylor et al., 2022) is trained on documents from a similar domain, we include it as part of zero-shot baselines along with the recent Falcon-180B (Almazrouei et al., 2023).

**Open and closed source models** We use different chat versions (7B, 13B, 70B) of Llama 2[12] (Touvron et al., 2023) as zero-shot open-source LLM baselines. For closed-source models, we evaluate zero-shot both GPT-3.5-turbo (Brown et al., 2020) and GPT-4 (OpenAI, 2023). Since they perform best in our initial evaluations, we use the closed-source models in combination with the different generation strategies (Per-cite, Sentence by sentence, plan-based, and learned plan) from 2.1.

## 5 RESULTS AND OBSERVATIONS

From Table 2, we first note that unsupervised extractive models provide a strong baseline compared to abstractive 0-shot single document summarization baselines. Fine-tuning these abstractive models on Multi-XScience (originally released with the benchmark) improves performance at least to the level of extractive models. We reproduce the PRIMER model using their open-source code but find

---

[6]Code will be released at `github.com`

[7]`https://huggingface.co/spaces/evaluate-metric/rouge` Since it is a known issue in the NLG community of different implementations producing different results, we stick to evaluate==0.4.0 for reporting all the results, reproducing the ROUGE scores for baselines from Multi-XScience model outputs.

[8]`https://spacy.io/usage/linguistic-features`

[9]`https://app.endpoints.anyscale.com/`

[10]`https://platform.openai.com/docs/guides/gpt`

[11]`https://github.com/huggingface/text-generation-inference`

[12]All Llama 2-Chat models herein referred to as Llama 2 and GPT-3.5-turbo as GPT-3.5 in text for brevity.

| Model Class | Model | ROUGE1 ↑ | ROUGE2 ↑ | ROUGEL ↑ |
|---|---|---|---|---|
| Extractive | One line baseline | 26.869 | 4.469 | 14.386 |
| | LexRank | 30.916 | 5.966 | 15.916 |
| | TextRank | 31.439 | 5.817 | 16.398 |
| Abstractive Finetuned | Hiersum | 29.861 | 5.029 | 16.429 |
| | Pointer-Generator | 33.947 | 6.754 | 18.203 |
| | PRIMER | 26.926 | 5.024 | 14.131 |
| Abstractive 0-shot | Long T5 | 19.515 | 3.361 | 12.201 |
| | Flan T5 | 21.959 | 3.992 | 12.778 |
| | Galactica-1.3B | 18.461 | 4.562 | 9.894 |
| | Falcon-180B | 22.876 | 2.818 | 12.087 |
| Open-source 0-shot | Llama 2-Chat 7B | 24.636 | 5.189 | 13.133 |
| | Llama 2-Chat 13B | 26.719 | 5.958 | 13.635 |
| | Llama 2-Chat 70B | 28.866 | 6.919 | 14.407 |
| Closed-source 2-stage | GPT-3.5-turbo (Per cite) 1st stage | 26.483 | 6.311 | 13.718 |
| | GPT-3.5-turbo (Per cite) 2nd stage | 24.359 | 5.594 | 12.859 |
| | GPT-3.5-turbo (Sentence by sentence) | 31.654 | 6.442 | 15.577 |
| Closed-source 0-shot | GPT-3.5-turbo (0-shot) | 29.696 | 7.325 | 14.562 |
| | GPT-4 (0-shot) | 33.213 | 7.609 | 15.798 |
| Plan | GPT-3.5-turbo (Learned plan) | 32.187 | 7.788 | 15.398 |
| | Llama 2-Chat 70B (Plan) | 34.654 | 8.371 | 17.089 |
| | GPT-3.5-turbo (Plan) | 35.042 | 8.423 | 17.136 |
| | GPT-4 (Plan) | **37.198** | **8.859** | **18.772** |

Table 2: Zero-shot and finetuned results for different models on the Multi-XScience dataset.

| Model | ROUGE1 ↑ | ROUGE2 ↑ | ROUGEL ↑ |
|---|---|---|---|
| CodeLlama 34B-Instruct | 22.608 | 5.032 | 12.553 |
| CodeLlama 34B-Instruct (Plan) | 27.369 | 5.829 | 14.705 |
| Llama 2-Chat 7B | 23.276 | 5.104 | 12.583 |
| Llama 2-Chat 13B | 23.998 | 5.472 | 12.923 |
| Llama 2-Chat 70B | 23.769 | 5.619 | 12.745 |
| GPT-3.5-turbo (0-shot) | 25.112 | 6.118 | 13.171 |
| GPT-4 (0-shot) | 29.289 | 6.479 | 15.048 |
| Llama 2-Chat 70B (Plan) | 30.919 | 7.079 | 15.991 |
| GPT-3.5-turbo (Plan) | 30.192 | 7.028 | 15.551 |
| GPT-4 (Plan) | **33.044** | **7.352** | **17.624** |

Table 3: Zero-shot results on the proposed MXS-2308 dataset.

results to be lower than reported. As such, we consider the Pointer-Generator method to be the current state-of-the-art (SOTA).

Single-document summarizers (LongT5, Flan T5) perform poorly in zero-shot settings with limited ability to cite references. We are limited in the prompt we can provide (because of the training prompts) and resort to 'Summarize the text and cite sources'. Galactica's performance is encouraging compared to other models in the same group, but inspecting its output reveals that it generates the whole introduction of the paper, instead of the related work. The model is very sensitive to the prompts used (mostly as suffixes) and struggles to follow instructions. Falcon 180-B, on the other hand, has the tendency to hallucinate user turns and considers this task as multiple turns of user-system exchange, even though we use prompt engineering to generate relevant outputs.

We find that all recent versions (7B,13B,70B) of zero-shot Llama 2 models underperform both the supervised Pointer-Generator baseline (with the exception of 70B on ROUGE2) and their GPT counterparts. All Llama 2 models have the tendency to produce output in bullet points and also provide references. We find that closed-sourced models like GPT-3.5-turbo and GPT-4 achieve SOTA in the zero-shot setting. However, the proposed sentence-by-sentence and per-citation strategies deteriorate the performance of GPT models.

| Model | Multi-XScience | | | MXS-2308 | | |
|---|---|---|---|---|---|---|
| | % ↑ | Mean ↓ | Max ↓ | % ↑ | Mean ↓ | Max ↓ |
| GPT-3.5-turbo (Plan) | 4.73 | 3.65 | 17 | 3 | 4.7 | 16 |
| Llama 2-Chat 70B (Plan) | 19.04 | 2.66 | 22 | 17.4 | 2.72 | 18 |
| GPT-4 (Plan) | 60.7 | -0.01 | 8 | 70.6 | -0.16 | 5 |

Table 4: We show % of responses with the same number of lines as the plan for both datasets. Here we also show the mean and max difference in lines generated by the model vs. the original plan. -ive implies that a model generated fewer lines than the plan. We find GPT-4 to follow the plan more closely compared to Llama 2 and GPT-3.5 which struggles to follow the exact details.

| Model | ROUGE1 ↑ | ROUGE2 ↑ | ROUGEL ↑ |
|---|---|---|---|
| StarCoder | 12.485 | 1.104 | 6.532 |
| Lemur-70B | 15.172 | 2.136 | 7.411 |
| CodeLlama 34B-Instruct | 25.482 | 5.814 | 13.573 |

Table 5: 0-shot results using code-based models on Multi-XScience dataset. CodeLlama performs reasonably well in generating natural language compared to the other code based counterparts.

Our teacher-forced plan-based framework improves the scores over the 0-shot baseline for both closed-sourced (GPT-3.5 and GPT-4) and open-sourced LLMs with Llama 2 70B achieving similar scores as GPT-3.5 on both the original Multi-XScience and the new MXS-2308 dataset (in Table 3). Llama 2 70B gets more uplift with the plan compared to GPT models where manual inspection reveals fewer hallucinations in the outputs (see qualitative results in Table 10 in Appendix using our abstract). In Table 4, we find that GPT-4 tends to follow the plan more closely. It follows the exact plan 60% of the time. Llama 2 70B comes in second place in following the plan instructions and GPT-3.5 struggles to follow the plan precisely. We also experiment with a learned plan strategy where the model first generates a plan and then autoregressively generates the output. Though it improves the results over 0-shot baseline, it does not outperform the teacher-forced plan generation.

There is a considerable drop in performance on MXS-2308 dataset compared to the original Multi-XScience in terms of ROUGE1/2. It gives more credibility to the hypothesis that the Multi-XScience test set is in the training data of these LLMs and/or that there is a shift in the distribution of these more recent papers. Nevertheless, we found a similar pattern of scores and ranking as observed for Multi-XScience.

**Code LLMs** We evaluate the performance of code-generating LLMs to write related-work section which requires more formal and structured language. Since Code LLMs are pre-trained on text they might offer the best of both worlds. However, we observe that for our task, the models produce bibtex and Python code with relevant comments as part of the generated outputs. As shown in Table 5, CodeLlama (34B Instruct) is good at following instructions and at generating natural language (ROUGE2 of 5.8 and 5.02 on Multi-XScience and MXS-2308 dataset). With a plan, CodeLlama even surpasses vanilla 0-shot Llama 2 70B (Table 3).

**Longer context and Llama 2 fine-tuning** While Llama 2 can ingest 4096 tokens, recent studies have found that it uses 19% more tokens (Kadous, 2023) than GPT-3.5 or GPT-4 (2048 and 4096 tokens respectively), implying that the effective number of words in Llama 2 is considerably lower than GPT-4 and only a bit higher than GPT-3.5. We experiment with the popular RoPE scaling (Su et al., 2021) in 0-shot Llama models to increase the context length (4k–6k). This permits using the full text of the papers instead of just their abstracts. Results in Table 6 show that directly using RoPE scaling on 0-shot models produces gibberish results. Instead, one needs to fine-tune the model with the longer context. In fact, a plan-based-longer-context CodeLlama (initialized from Llama 2 and trained with a 16k token context through RoPE scaling) improves on ROUGE1/L, but achieves comparable results as a shorter-context plan-based CodeLlama on ROUGE2.

In parallel, we also fine-tune Llama 2 models on the train set with the original shorter context, but they are very sensitive to hyperparameter configuration. When we instruct-finetune Llama 2 7B, it initially produces code. We find a slight improvement when fine-tuning the Llama 2 7B model for 30k steps with an LR of 5e-6 over 0-shot model (see Table 7), but it quickly overfits as we increase

| Model | ROUGE1 ↑ | ROUGE2 ↑ | ROUGEL ↑ |
|---|---|---|---|
| Llama 2-Chat 7B (4000 words) | 17.844 | 1.835 | 10.149 |
| Llama 2-Chat 7B (5000 words) | 17.254 | 1.736 | 9.986 |
| Llama 2-Chat 7B (6000 words) | 17.179 | 1.647 | 9.897 |
| Llama 2-Chat 13B (4000 words) | 20.071 | 3.516 | 10.916 |
| Llama 2-Chat 13B (5000 words) | 20.722 | 3.714 | 11.13 |
| Llama 2-Chat 13B (6000 words) | 17.179 | 1.647 | 9.897 |
| Llama 2-Chat 70 (4000 words) | 19.916 | 2.741 | 10.456 |
| Llama 2-Chat 70B (5000 words) | 19.675 | 2.605 | 10.48 |
| Llama 2-Chat 70B (6000 words) | 20.437 | 2.976 | 10.756 |
| CodeLlama 34B-Instruct (4000 words) | 27.425 | 5.815 | 14.744 |

Table 6: Zero-shot results using RoPE scaling for larger context on MXS-2308 dataset. Here we report the max number of words used for truncation instead of the tokens.

| Model | ROUGE1 ↑ | ROUGE2 ↑ | ROUGEL ↑ |
|---|---|---|---|
| Llama 2-Chat 7B - 0-shot | 26.719 | 5.958 | 13.635 |
| Llama 2-Chat 7B - 10k steps (LR 5e-6) | 24.789 | 5.986 | 12.708 |
| Llama 2-Chat 7B - 30k steps (LR 5e-6) | 27.795 | **6.601** | 14.409 |
| Llama 2-Chat 7B - 60k steps (LR 1e-5) | 22.555 | 5.511 | 11.749 |

Table 7: Results after fine-tuning Llama 2-Chat 7B on Multi-XScience dataset

the LR or the number of steps. We leave hyperparameter optimization, the fine-tuning of larger models with RoPE scaling and plan-based generation for future work.

**Coverage, human and automatic LLM evaluation** We evaluate coverage as the percentage of model outputs with the same number of citations as ground truth (identified using regex on the delexicalized citation in the generated related work). Table 8 shows the efficacy of plan-based approaches. All plan models provide more coverage than their counterparts with GPT-4 achieving 98% in covering all the citations. The largest uplift is for (vanilla) 0-shot Llama 2 70B. Using a plan raises its coverage from 59% to 82%. In line with results in Table 4, we find that coverage of GPT-3.5 does not improve much. It struggles to provide the same number of sentences as the plan.

Similar to Zheng et al. (2023), we use GPT-4 as a judge to get automatic LLM results (prompts in Appendix, Figures 8 and 9). We use both pairwise comparison and single answer grading, both using GT-related work as reference.[13] We find that GPT-4 provides similar grades to the results of all the models with the exception of 0-shot GPT-4. Manual inspection of the judge's review[14] suggests that while the judge finds output of plan-based methods 'to be more concise, it finds the output of 0-shot models to be more detailed and comprehensive description of the related work'. Table 8 also shows a similar pattern for average win rate. We note that on average the GPT-4 without a plan generate significantly longer passages compared to GPT-4 with a plan (215 words versus 125).

The above results seem to contradict previous evaluations using the ROUGE metric (e.g. Table 2). There are a few caveats. First, we note that there is no evidence that the judge's scores correlate with human judgments, but that this automated evaluation is becoming standard nonetheless (Dettmers et al., 2023). Further, the judge's justification hints that authors could have personal preferences rendering the comparison of these two models subjective (e.g., some authors could prefer concision to comprehensiveness and vice versa). A preliminary human evaluation supports this. It finds a low inter-rater agreement between the passages generated by GPT-4 with and without a plan. This is true even when using specific instructions describing the qualities of a good related-works passage. In practice, authors have different writing styles and their perception of the importance of certain ma-

---

[13]Since we are limited by max tokens, we do not include cited references as input to the judge.

[14]An excerpt of review: While Assistant 1's output is more concise, Assistant 2 provides a more in-depth and comprehensive analysis of the related works. Therefore, Assistant 2's output is more detailed and comprehensive description of the related work, which might be beneficial for a reader unfamiliar with this field.

| Model | Avg Rating ↑ | Avg Win Rate ↑ | Coverage ↑ | Avg. words |
|-------|--------------|----------------|------------|------------|
| Llama 2-Chat 70B (0-shot) | 6.197 | 33.4% | 59.31% | 284.65 |
| Llama 2-Chat 70B (Plan) | 6.321 | 32.6% | 82.62% | 191.45 |
| GPT-3.5-turbo (0-shot) | 6.489 | 56.4% | 63.11% | 293.69 |
| GPT-3.5-turbo (Plan) | 6.459 | 39.2% | 68.03% | 202.81 |
| GPT-4 (0-shot) | 7.428 | 77.6% | 91.34% | 215.15 |
| GPT-4 (Plan) | 6.763 | 30.4% | **98.52%** | 125.1 |

Table 8: Coverage and automatic LLM evaluation on the Multi-XScience dataset

terials also vary (so do space constraints). These observations highlight the need to design systems with humans-in-the-loop. We believe our plan-based approach could serve as a good basis for this. We think that more nuanced human-evaluation protocols could also help one understand the ways in which these tools can best be used.

# 6 CONCLUSION

In this work, we present a comprehensive study of using LLMs for the task of multi-document summarization where we explore the zero-shot abilities of recent large language models (LLMs). We propose and examine a novel plan based strategy strategy for literature review generation with LLMs improving upon zero-shot vanilla generation and achieving SOTA on Multi-XScience dataset. We also extend the Multi-XScience corpus – which previously only included the abstracts of papers to include the full text of research papers. While we didn't find any improvements incorporating more content with current shorter context LLMs, we expect the results to improve with the release of new LLMs having an expanded context window. Furthermore, we also propose a new challenging test corpus and provide a SOTA benchmark with GPT-4 plan-based generation.

**Broader Impact** The potential for LLM and other NLP technology to help in scientific writing has led to the emergence of systems such as Explainpaper which helps researchers understand the contents of the paper and Writefull[15] for title and abstract generation. Scite[16] helps find appropriate resources and references while writing research papers. The usage of LLMs by researchers is so evident that some conferences (like ICLR) collect statistics from authors about their usage of these tools for retrieval (or discovery) or paraphrasing the related work section and provide LLM-specific guidelines to authors.[17] Given the growing impact of LLM-based writing assistants, we are optimistic that our plan-based approach may help improve the quality of text generated as an aid for creating a related work section of a paper. Given the problems that prior methods have had with citation hallucination, we believe that citation-conditioned LLMs have significant potential to improve the utility and quality of generated text.

**Limitations and future work** Even though we manually find a reduction in cases of hallucinations with our plan strategy, we plan to run a proper study to quantify it in the future. While we explore the application of LLMs for academic literature, our work is limited to using the static Multi-XScience dataset. This strategy also assumes that we already have filtered relevant papers corresponding to the main paper. In real life, we might want a system to recommend the relevant papers for the work automatically.

Our work also raises an important question as to how a literature review or related work section is written and how we can capture different styles in which humans write a cohesive storyline demonstrating key differences and insights of the previous work compared to different approaches they propose. Hence just a summary, even if somehow conditioned on context, is wholly inadequate. We consider this as an open-ended research question and invite fellow researchers in the community for further discussion.

---

[15] https://www.explainpaper.com/, https://x.writefull.com/

[16] https://scite.ai/

[17] ICLR'24 Large Language Models guidelines https://iclr.cc/Conferences/2024/CallFor Papers

## ETHICS STATEMENT

While writing assistant technology could have great promise as an aide to scientists, we think their use should be disclosed to the reader. As such assistants become more powerful they might be abused in certain contexts, for example where students are supposed to create a literature review as a part of their learning process. The use of such tools might also be problematic as authors of scientific work should read the articles that they cite and heavy reliance on such tools could lead to short term gains at the cost of a deeper understanding of a subject over the longer term. Any commercially deployed or systems actually used by authors should also contain appropriate mechanisms to detect if words have been copied exactly from the source material, and provide that content in a quoted style.

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

# A  APPENDIX

| |
|---|
| **Abstract of Multi-XScience paper (Lu et al., 2020)** |
| **Reference @cite_1:** Multi-document summarization is a challenging task for which there exists little large-scale datasets. We propose Multi-XScience, a large-scale multi-document summarization dataset created from scientific articles. MultiXScience introduces a challenging multi-document summarization task: writing the related-work section of a paper based on its abstract and the articles it references. Our work is inspired by extreme summarization, a dataset construction protocol that favours abstractive modeling approaches. Descriptive statistics and empirical results—using several state-of-the-art models trained on the MultiXScience dataset—reveal that Multi-XScience is well suited for abstractive models. |
| **Abstract of Extractive and Abstractive Summarization paper (Pilault et al., 2020)** |
| **Reference @cite_2:** We present a method to produce abstractive summaries of long documents that exceed several thousand words via neural abstractive summarization. We perform a simple extractive step before generating a summary, which is then used to condition the transformer language model on relevant information before being tasked with generating a summary. We show that this extractive step significantly improves summarization results. We also show that this approach produces more abstractive summaries compared to prior work that employs a copy mechanism while still achieving higher rouge scores. Note: The abstract above was not written by the authors, it was generated by one of the models presented in this paper. |
| **Abstract of Galactica paper (Taylor et al., 2022)** |
| **Reference @cite_3:** Information overload is a major obstacle to scientific progress. The explosive growth in scientific literature and data has made it ever harder to discover useful insights in a large mass of information. Today scientific knowledge is accessed through search engines, but they are unable to organize scientific knowledge alone. In this paper we introduce Galactica: a large language model that can store, combine and reason about scientific knowledge. We train on a large scientific corpus of papers, reference material, knowledge bases and many other sources. We outperform existing models on a range of scientific tasks. On technical knowledge probes such as LaTeX equations, Galactica outperforms the latest GPT-3 by 68.2% versus 49.0%. Galactica also performs well on reasoning, outperforming Chinchilla on mathematical MMLU by 41.3% to 35.7%, and PaLM 540B on MATH with a score of 20.4% versus 8.8%. It also sets a new state-of-the-art on downstream tasks such as PubMedQA and MedMCQA dev of 77.6% and 52.9%. And despite not being trained on a general corpus, Galactica outperforms BLOOM and OPT-175B on BIG-bench. We believe these results demonstrate the potential for language models as a new interface for science. We open source the model for the benefit of the scientific community. |
| **Plan for Table 10** |
| Please generate 5 sentences in 120 words. Cite @cite_1 at line 1, 3 and 5. |
| **Plan for Table 10** |
| Please generate 5 sentences in 120 words. Cite @cite_1 at line 1 and 3. Cite @cite_2 at line 2 and 5. Cite @cite_3 at line 4 and 5. |

Table 9: Abstracts of papers which are reference citations in Tables 10 and 11.

**GPT-4 0-shot**

The field of multi-document summarization has been a challenging one due to the lack of large-scale datasets (@cite_1). One significant attempt to address this gap is the Multi-XScience dataset, a large-scale collection of scientific articles designed to simulate the task of writing the related-work section of a research paper (@cite_1). This approach was inspired by extreme summarization, a methodology that promotes abstractive modeling approaches. However, the current study offers a novel strategy for literature review generation using large language models (LLMs), which goes beyond the scope of the Multi-XScience dataset. The proposed method involves the creation of an intermediate plan, which is then used to generate the actual text. Empirical evidence suggests that this approach improves the quality of the generated literature reviews compared to the conventional zero-shot generation. Additionally, this work expands the Multi-XScience corpus to include the full text of research papers, further enhancing the depth and breadth of information available for literature review generation. This study also ensures the validity of zero-shot experiments by creating a new test corpus from recent arXiv papers, thus avoiding test set contamination. This reflects a more comprehensive and rigorous approach to the generation of literature reviews using LLMs.

**GPT-3.5 Plan**

The task of generating literature reviews using large language models (LLMs) has gained attention recently. One approach is to condition the LLM on the abstracts of the papers to be cited and generate the literature review directly. However, the authors propose a novel strategy where they first generate a plan for the review and then use it to generate the actual text. This intermediate planning step is found to improve the quality of the generated literature reviews compared to vanilla zero-shot generation. The authors extend the Multi-XScience corpus, previously containing only abstracts, to include the full text of research papers. Additionally, they create a new test corpus of recent arXiv papers, ensuring that their zero-shot experiments are not affected by contamination from the test set. The Multi-XScience dataset, used in their work, is particularly suitable for abstractive models due to its challenging multi-document summarization task (@cite_1). The empirical results from several state-of-the-art models trained on this dataset demonstrate its effectiveness for abstractive modeling approaches.

**GPT-3.5 0-shot**

Prior work in the field of generating literature reviews has primarily focused on the use of large language models (LLMs) and multi-document summarization. Multi-document summarization is a challenging task due to the lack of large-scale datasets (@cite_1). The Multi-XScience dataset was proposed as a solution to this problem, specifically designed for generating the related-work section of a paper based on its abstract and the articles it references. This dataset introduced a challenging multi-document summarization task that favors abstractive modeling approaches (@cite_1).

Our work builds upon these prior efforts by exploring the zero-shot abilities of recent LLMs for the task of generating literature reviews. We propose a novel strategy that involves generating an intermediate plan for the review, which is then used to generate the actual text. While modern LLMs can generate literature reviews without such intermediate plans, our empirical study shows that the use of these plans improves the quality of the generated reviews compared to vanilla zero-shot generation.

In addition to extending the Multi-XScience corpus to include the full text of research papers, we also create a new test corpus consisting of recent arXiv papers. This allows us to ensure that our zero-shot experiments are not affected by test set contamination.

The prior work on multi-document summarization and the Multi-XScience dataset provides a valuable foundation for our study. However, they do not specifically address the use of intermediate plans for literature review generation, which is a key contribution of our work. By incorporating these intermediate plans, we aim to enhance the quality and coherence of the generated literature reviews.

**Llama 2 Plan**

The task of generating literature reviews for scientific research papers has gained significant attention in recent years, with a focus on using large language models (LLMs) for zero-shot generation. Inspired by extreme summarization, Multi-XScience was introduced as a large-scale dataset for multi-document summarization, including the challenging task of writing related-work sections based on abstracts and referenced articles. Our work builds upon this foundation by exploring the effectiveness of intermediate plans for literature review generation. Our empirical study shows that these plans improve the quality of generated literature reviews, and we contribute to the extension of the Multi-XScience corpus to include full-text papers. Additionally, we create a new test corpus to ensure zero-shot experiments are free from test set contamination. (@cite_1)

**Llama 2 0-shot**

In recent years, there has been growing interest in using large language models (LLMs) for zero-shot literature review generation. However, most studies have relied on training or prompting LLMs to condition on all abstracts of papers to be cited, which can be limiting in terms of quality and efficiency. In contrast, our proposed approach involves generating a plan for the literature review and then using it to generate the actual text. We evaluate the effectiveness of this approach using a novel extended version of the Multi-XScience corpus, which now includes the full text of research papers. Our results show that intermediate plans improve the quality of generated literature reviews over vanilla zero-shot generation. One of the key challenges in multi-document summarization is the lack of large-scale datasets. To address this, we have created a new test corpus consisting of recent arXiv papers with full content, ensuring that our zero-shot experiments are not contaminated by the release of the LLMs used in our study. Our work builds on the concept of extreme summarization, which favors abstractive modeling approaches. We have employed several state-of-the-art models trained on the Multi-XScience dataset to evaluate the effectiveness of our approach. The results show that our dataset is well-suited for abstractive models, providing a valuable resource for researchers working in this area. In summary, our contribution includes a novel approach to literature review generation using LLMs, an extended version of the Multi-XScience corpus, and a new test corpus. Our results demonstrate the effectiveness of our approach and highlight the potential of using LLMs for zero-shot literature review generation. References: * @cite_1: Multi-XScience: A Large-Scale Multi-Document Summarization Dataset. * @cite_2: Extreme Summarization: A Dataset Construction Protocol for Abstractive Modeling.

Table 10: We show outputs of different models corresponding to the same plan and prompt as paragraph of our related work (by GPT-4 Plan). Llama 2 70B 0-shot has the tendency to hallucinate citation (in red) which is remedied in plan based generation. Abstract of reference paper shown in Table 9 while the main abstract is a slight modified version of our current abstract.

**GPT-4 Plan**

The concept of generating the related-work section from abstracts and referenced articles is not entirely new, as evidenced by the Multi-XScience dataset proposed by @cite_1. The approach of extending abstractive summarization to lengthy documents via an extractive step for conditioning, as described by @cite_2, aligns with the proposed methodology here. However, the strategy introduced in this study differentiates itself by creating an iterative plan prior to the generation of the review, a step not present in the work of @cite_1. The LLM used in this study also builds upon the capabilities of models like Galactica @cite_3, which have demonstrated exceptional performance on scientific tasks. The results of @cite_2 and @cite_3 support the potential of large language models in scientific literature review generation, while the proposed approach aims to further enhance the quality of these outputs through structured planning.

**GPT-4 0-shot**

The literature review generation task, a form of multi-document summarization, has been tackled through different strategies. @cite_1 proposed a dataset, MultiXScience, to foster abstractive approaches, while @cite_2 used an extractive step to enhance neural summarization. Our work complements these by introducing intermediate plans to improve generation quality. Meanwhile, @cite_3's Galactica demonstrated potential for language models in scientific tasks, a direction we explore with zero-shot experiments.

**GPT-3.5 Plan**

The task of generating a literature review using large language models (LLMs) has gained attention recently. One approach is to condition the LLM on the abstract and content of related papers to generate the review (@cite_1). However, it has been observed that generating an intermediate plan for the review improves its quality compared to vanilla zero-shot generation. This motivated the proposal and examination of a novel strategy where a plan is first generated and then used to generate the actual text (@cite_1). Another method to improve abstractive summarization is by performing an extractive step before generating the summary. This step conditions the transformer language model on relevant information, leading to better summarization results compared to prior work that employs a copy mechanism (@cite_2). Additionally, the use of large language models like Galactica has shown promising results in organizing and reasoning about scientific knowledge (@cite_3). Galactica outperforms existing models on technical knowledge probes and downstream tasks, demonstrating the potential for language models as a new interface for science (@cite_3).

**GPT-3.5 0-shot**

Prior work has explored the use of large language models (LLMs) for various natural language processing tasks. Multi-document summarization is one such task, and the Multi-XScience dataset (@cite_1) has been created specifically for this purpose. This dataset focuses on writing the related-work section of a scientific paper based on its abstract and the articles it references. The dataset construction protocol favors abstractive modeling approaches, making it suitable for LLMs. Another approach to abstractive summarization is presented in a paper (@cite_2), where the authors propose a method to generate summaries of long documents. They employ an extractive step to gather relevant information before conditioning the transformer language model for summary generation. This approach yields more abstractive summaries compared to prior work, while still achieving higher rouge scores. Furthermore, the concept of using LLMs for scientific knowledge organization and reasoning is introduced in the Galactica model (@cite_3). This model is trained on a large scientific corpus and outperforms existing models on various scientific tasks. It demonstrates superior performance on technical knowledge probes and reasoning tasks, surpassing GPT-3 and other models. Galactica also achieves state-of-the-art results on downstream tasks such as PubMedQA and MedMCQA, showcasing the potential of LLMs as a new interface for science. Based on these prior works, we propose a novel strategy for literature review generation using LLMs. We first generate a plan for the review and then use it to generate the actual text. While modern LLMs can generate literature reviews without intermediate plans, our empirical study indicates that these plans improve the quality of generated reviews. Additionally, we create a new test corpus consisting of recent arXiv papers, ensuring that our zero-shot experiments are not affected by test set contamination.

**Llama 2 Plan**

In recent years, there has been a growing interest in using large language models (LLMs) for scientific literature review generation (@cite_1). One approach is to use a novel strategy that involves generating a plan for the review and then using it to generate the actual text (@cite_1). This approach has been shown to improve the quality of generated literature reviews over vanilla zero-shot generation (@cite_1). Another approach is to use a simple extractive step before generating a summary, which is then used to condition the transformer language model on relevant information (@cite_2). This approach has been shown to produce more abstractive summaries compared to prior work that employs a copy mechanism while still achieving higher ROUGE scores (@cite_2). Additionally, there have been efforts to train LLMs on large scientific corpora to improve their ability to store, combine, and reason about scientific knowledge (@cite_3). For example, Galactica, a large language model, has been trained on a range of scientific tasks and has outperformed existing models on technical knowledge probes, reasoning, and downstream tasks (@cite_3). These advancements in LLMs have the potential to revolutionize the way we interact with scientific literature and accelerate scientific progress.

**Llama 2 0-shot**

In this section, we discuss the related work in the field of scientific literature review generation using large language models (LLMs). Recent works have explored the use of LLMs for multi-document summarization tasks, such as generating the related work section of a paper based on its abstract and the articles it references (@cite_1). These works have shown that LLMs can produce high-quality summaries, but may suffer from the lack of relevant information and context (@cite_2). To address this issue, we propose a novel strategy for literature review generation that involves generating a plan for the review before generating the actual text. This approach has been shown to improve the quality of generated literature reviews over vanilla zero-shot generation (@cite_3). Additionally, we create a new test corpus consisting of recent arXiv papers to ensure that our zero-shot experiments do not suffer from test set contamination. Our work builds upon the recent advances in LLMs and their ability to store, combine, and reason about scientific knowledge (@cite_3). We believe that our approach has the potential to revolutionize the way scientific literature is reviewed and organized, and we open source our model for the benefit of the scientific community. In summary, our related work demonstrates the potential of LLMs for scientific literature review generation, and highlights the importance of context and information in producing high-quality summaries. Our proposed approach aims to address these limitations and provide a more effective way of generating literature reviews using LLMs.

Table 11: We show outputs of different models corresponding to the reference cited abstracts and plan mentioned in Table 9 while the main abstract is a slight modified version of our current abstract. In this example, though, we have all the citations covered by all the models, we can see GPT-4 (Plan) output to be concise and closely following the plan.

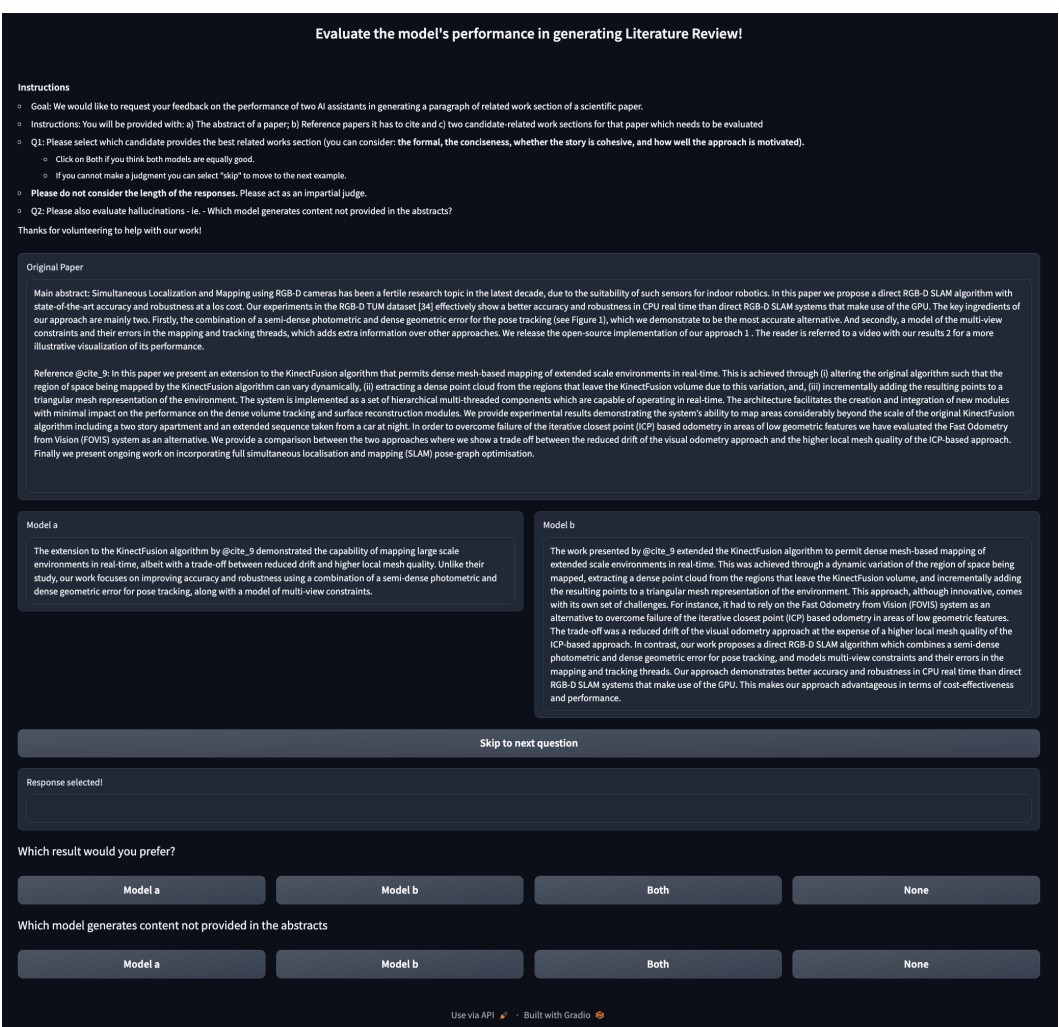

Figure 2: Interface of our human evaluation setup.

---

### Plan based prompt

You will be provided with an abstract of a scientific document and other references papers in triple quotes. Your task is to write the related work section of the document using only the provided abstracts and other references papers. Please write the related work section creating a cohesive storyline by doing a critical analysis of prior work comparing the strengths and weaknesses while also motivating the proposed approach. You are also provided a plan mentioning the total number of lines and the citations to refer in different lines. You should cite the other related documents as (@cite_#) whenever it is referred in the related work. Do not cite abstract. Do not include any extra notes or newline characters at the end. Do not copy the abstracts of reference papers directly but compare and contrast to the main work concisely. Do not provide the output in bullet points. Do not provide references at the end. Please follow the plan when generating sentences, especially the number of lines to generate. Provide the output in max 200 words.

Suffix - "Related Work: \n"

---

Figure 3: Prompt used for Plan based generation.

---

### Vanilla 0-shot prompt

You will be provided with an abstract of a scientific document and other references papers in triple quotes. Your task is to write the related work section of the document using only the provided abstracts and other references papers. Please write the related work section creating a cohesive storyline by doing a critical analysis of prior work comparing the strengths and weaknesses while also motivating the proposed approach. You should cite the other related documents as (@cite_#) whenever it is referred in the related work. Do not cite abstract. Do not include any extra notes or newline characters at the end. Do not copy the abstracts of reference papers directly but compare and contrast to the main work concisely. Do not provide the output in bullet points. Do not provide references at the end. Provide the output in max 200 words.

Suffix - "Related Work: \n"

---

Figure 4: Prompt used for Vanilla 0-shot generation.

---

### Learned plan prompt

You will be provided with an abstract of a scientific document and other reference papers in triple quotes. Your task is to write the related work section of the document using only the provided abstracts and other reference papers. Please generate the related work creating a cohesive storyline by doing a critical analysis of prior work comparing the strengths and weaknesses while also motivating the new work. You should cite the other related documents as (@cite_#) whenever it is referred to in the related work. Do not cite abstract. Do not include any extra notes or newline characters at the end. Do not copy the abstracts of reference papers directly but compare and contrast to the main work concisely. Do not provide the output in bullet points. Do not provide references at the end. Provide the output in max 200 words. You should first generate a plan, mentioning the total number of lines, words and the citations to refer to in different lines. You should follow this plan when generating sentences. \n Example: \n\n Plan: Generate the related work in [number] lines using max [number] words. Cite @cite_# on line [number]. Cite @cite_# on line [number].\n

Suffix - "Related Work: \n"

Figure 5: Prompt used when plan is learned during generation.

---

### Sentence by sentence prompt

You are assisting a researcher to write a related work section of a paper sentence by sentence. You will be provided with an abstract of the scientific document and raw draft of generated related work till now in triple quotes. Additionally, you will be provided with a reference paper if it has to be cited in the sentence. Your task is to write another 1 sentence for the related work section of the document or paraphrase the draft using only the abstract and other reference papers if provided. Initially, the raw draft would be empty. Please complete the related work creating a cohesive storyline by doing a critical analysis of prior work comparing the strengths and weaknesses while also motivating the proposed approach. You should cite the other related documents as (@cite_#) only whenever it is referred to in the related work. Do not cite abstract. Do not include any extra notes or newline characters at the end. Do not copy the abstracts of reference papers directly but compare and contrast to the main work concisely. Do not provide the output in bullet points. Do not provide references at the end. Provide the output in max 200 words. Provide the complete related work including the new sentence.

Suffix - "Related Work: \n"

Figure 6: Prompt used for Sentence by sentence generation.

## Per cite prompt

You will be provided with an abstract of a scientific document and other references paper in triple quotes. Your task is to write the related work section of the document using only the provided abstracts and other references papers. Please generate the related work creating a cohesive storyline by doing a critical analysis of prior work comparing the strengths and weaknesses while also motivating the new work. You should cite the other related documents as (@cite_#) whenever it is referred in the related work, comparing with the main paper. Do not cite abstract. Do not provide references at the end. Provide the output in 1-2 sentence.

Suffix - "Related Work: \n"

Figure 7: Prompt used for generating output per citation.

## Single evaluation prompt

Please act as an impartial judge and evaluate the quality of the response provided by an AI assistant in generating a paragraph of scientific related work section of a paper. You will also be provided with the original related work and abstract of the paper in triple quotes. For evaluation you can consider if the output is formal, concise, whether the story is cohesive, and how well the approach is motivated. Your evaluation should also consider that all the references present in original related work are covered in the output and penalize if the citation is missing. Do not allow the length of the responses to influence your evaluation. Be as objective as possible. Begin your evaluation by comparing the response with the original related work and provide a short explanation avoiding any potential bias. After providing your explanation, you must rate the response on a scale of 1 to 10. Do not output anything else other than the number in the last line. Remember, the output should be in the format:\n Review: \n\n Rating:

Figure 8: Prompt used for single reference based automatic evaluation using LLM.



### Pairwise evaluation prompt

We would like to request your feedback on the performance of two AI assistants in generating a paragraph of scientific related work section for a paper.\n You will also be provided with the original related work and abstract of the paper in triple quotes. Please select which candidate provides the best related works section (you can consider: the formal, the conciseness, whether the story is cohesive, and how well the approach is motivated). Your evaluation should also consider that all the references present in original related work are covered in the outputs of two assistants and penalize if the citation is missing. Do not allow the length of the responses to influence your evaluation. Please act as an impartial judge. Do not favor certain names of the assistants and ensure that the order of the responses does not affect your judgment. Be as objective as possible. Begin your evaluation by comparing the two responses and provide a short explanation avoiding any potential bias. Please remember that length of the responses is not a criteria for evaluation. \nOnce you carefully review both submissions, in a new line, choose the best answer between the answers of Assistant 1 and Assistant 2 by outputting the number as rating of 1 or 2 respectively, or choose 3 if the two assistants are equivalent. Do not output anything else other than the number in the last line. Remember, the output should be in the format:\n Review: \n\n Rating:



Figure 9: Prompt used for used for pairwise automatic evaluation using LLM.

