# OpenReview forum: "Plan-based Prompting Improves Literature Review Generation"
_ICLR.cc/2024/Conference — Submitted to ICLR 2024_

### Official Review · Reviewer_CcYt · 2023-10-30

**Soundness:** 2 fair
**Presentation:** 2 fair
**Contribution:** 2 fair
**Rating:** 3
**Confidence:** 3

**Summary:**

This paper aims to explore the task of literature generation.
To do this, the authors propose several plan-based prompting methods for LLMs, and assemble a new dataset for evaluation.
Although the initial motivation is appreciated, this paper is very difficult to follow, in particular for the experiment part, and many of the detailed are confusing.
Moreover, the plan prompting method is widely used for text generation, while this paper highly relies on human designed prompts.
Thus the novelty and generalization of the proposed method is somehow limited.

**Strengths:**

The problem that this paper aims to address is indeed important and can be useful in real-world applications.

**Weaknesses:**

1. The presentation of this paper is very poor and many details are missing. In particular, for section2, the authors provide multiple prompt methods. But it is very confusing that how models are asked to generate such plans. My understanding is that some of them require model training (But the authors claim that they are doing the zero-shot evaluation) while some are human crafted. And this part is not well aligned with the experiment section, I cannot fully understand the experimental setup and details. I suggest the authors make those sections more clear and well-structured.
2. Planning based method for text generation has been widely studied. I have listed a few below, but I believe there are more. And many of them have explored automatic planning generation. It seems that this paper highly replies on human designed prompting for planning, and some plans even require human writing (learned plan). So the novelty of this paper is very limited.
3. Experiments are difficult to follow as well. The authors propose 4 prompting methods  (Per-cite, Sentence by sentence, plan-based, and learned plan), but in some experimental results, they only show part of the results.
For example, in table 2, they experiment 4 methods for Turbo, but only 1 for GPT-4. And what does the "GPT-4 (Plan)" mean? does it refer to "GPT-4 plan-based"? Overall, it is very difficult to follow and compare their results.



LLM Blueprint: Enabling Text-to-Image Generation with Complex and Detailed Prompts

Dynamic Planning with a LLM

Skeleton-of-Thought: Large Language Models Can Do Parallel Decoding

Self-planning Code Generation with Large Language Models

**Questions:**

Please see weakness.

---

### Official Review · Reviewer_N3Vy · 2023-10-31

**Soundness:** 3 good
**Presentation:** 3 good
**Contribution:** 3 good
**Rating:** 5
**Confidence:** 3

**Summary:**

This paper proposes a plan-based literature review generation method for LLMs. It first generates a plan for the review and then uses the plan to generate the literature review conditional on the abstracts of papers to be cited. It also creates new arXiv datasets collected from the latest research paper to avoid contamination. The experimental results show that the intermediate plans can improve the zero-shot performance of LLMs for literature review generation.

**Strengths:**

* This paper investigates the zero-shot capability of recent LLMs to generate literature views. It further uses an intermediate plan to improve the performance of LLMs.
* The authors conduct comprehensive comparisons for extractive models, abstractive finetuned models, and zero-shot LLMs under different settings. They also investigate the performance of Code-generating LLMs (CodeLLaMA, StarCoder, Lemur-70B), LLaMA-2 with longer context, and finetuned LLaMA-2 on this task.

**Weaknesses:**

* The plan-based method requires manually designing a plan based on the ground truth in advance, which is unrealistic in real-world scenarios. The learned plan methods are not comparable to the methods with pre-defined plans based on Table 2. It indicates that the proposed method may be difficult to generalize to a new dataset without the ground truth summary.
* The novelty of the proposed method is limited. The most effective part is the manually designed plan. Based on that, they should also discuss some plan-based /outline-based prompt studies, such as [1-3].
* Some experimental details are missing. For example,
   * The detailed information of the proposed new data sets. For example, the size, average document length, average summary length, average citation number, training/validation/testing split, and so on.
   * How to choose the X (sentence number) and Y (words) in plan-based methods?
   * What generation configuration is used in LLaMA-2, ChatGPT-3.5, and GPT-4? For example, the greedy decoding or the sampling with temperature.
  * How was the human evaluation (In Section 5) conducted? The number of annotators, the inner agreement among annotators, the average compensation, the working hours, and the procedure of annotation should be described in detail.
   * How are Avg Rating, Avg Win Rate, and Coverage in Table 8 calculated?


[1] Re3: Generating Longer Stories With Recursive Reprompting and Revision
[2] Plan-and-Solve Prompting: Improving Zero-Shot Chain-of-Thought Reasoning by Large Language Models
[3] Self-planning Code Generation with Large Language Models

**Questions:**

See weaknesses above

---

### Official Review · Reviewer_ViNE · 2023-11-04

**Soundness:** 2 fair
**Presentation:** 1 poor
**Contribution:** 2 fair
**Rating:** 3
**Confidence:** 3

**Summary:**

This paper introduces a "plan-based" method for generating literature reviews, adding an intermediate planning step before the creation of the actual text. This approach is shown to enhance the quality of the generated reviews. Furthermore, the authors have developed a new test corpus, free from data contamination, to evaluate the performance of various models.

**Strengths:**

1. **Useful Dataset** - This paper constructs two new datasets for evaluating literature review generation. The authors promise to release these datasets, which could be of benefit to the community for future research endeavors.
2. **Comprehensive Investigation** - The study investigates a broad coverage of models, including fine-tuned extractive and abstractive models, open-source LLMs, code LLMs, and GPT-* models. Furthermore, the authors have developed variants of LLM-2 with different context lengths to examine the impact of longer contexts on generation quality.

**Weaknesses:**

1. **Lack of Details in Dataset Construction** - The detailed dataset statistics (e.g., average length of reference text, average number of papers cited) of the newly-constructed MXS-2308 are missing. Moreover, the authors should at least include one example from the constructed dataset for the case study, so that the readers could have better assessment of the proposed dataset and its challenges.
2. **Potential Leakage of Ground Truth Information in Plan-based Generation** - If I understand correctly, the "Plan-based" method proposed in this paper derives plans from the ground-truth text using a rule-based approach. This process could potentially lead to leakage of ground-truth information, resulting in generated texts that closely mirror the organization of the ground truth and, consequently, attain artificially high ROUGE scores.
3. **Limitations of Performance Evaluation**
   - The paper asserts that the proposed plan-based method can reduce hallucination in the generated output. However, the manuscript lacks both quantitative and qualitative evidence from human evaluations or automated metrics to substantiate this claim.
   - While the paper employs general-level automated evaluation systems like ROUGE and GPT judgment (without cited references as input), incorporating additional metrics that specifically assess the faithfulness of the generated text could strengthen the evaluation framework.
   - The setting and results of human evaluation is absent

**Questions:**

1. In real-world scenarios, the 'Related Work' section of a paper often contains numerous references, with many not discussed in detail. Moreover, multiple references may be cited within a single sentence. How did you handle these common cases during both the data construction and plan-generation stages?

2. For "Plan-based" approach, how do you decide on the exact parameters (number of sentences, word count, and citation placement) for each instance within your constructed dataset? (related to weakness 2)
> Please generate {num sentences} sentences in {num words} words. Cite {cite x} at line {line x}. Cite {cite y} at line {line y}.

---

### Official Review · Reviewer_u18m · 2023-11-06

**Soundness:** 3 good
**Presentation:** 3 good
**Contribution:** 2 fair
**Rating:** 3
**Confidence:** 4

**Summary:**

The paper explores the zero-shot abilities of LLMs in the scientific research paper review generation tasks. The paper reformulates the generation with a writing-plan-based intermediate task to improve the generation quality. The paper reviews several existing related work datasets and proposes a new MXS-2308 to avoid overlap with training data. The paper benchmarks several SOTA baselines including extractive baselines, abstraction fine-tuned baselines, abstraction zero-shot baselines, and open and closed source models. The plan method achieves significantly better performance compared to the others. The paper conducts both automatic and human evaluations.

**Strengths:**

1. The paper proposes a new writing-plan-based intermediate task for literature review generation. The paper prompts the model to generate a plan for reference sentences. The input consists of the query paper abstract and reference paper abstracts. To ensure papers in the dataset are not included in the pretraining set, the paper creates a new dataset, MXS-2308.
2. The paper conducts experiments on both Multi-XScience and MXS-2308. The paper benchmarks several SOTA baselines, including extractive baselines, abstraction fine-tuned baselines, abstraction zero-shot baselines, and open and closed source models. The paper conducts both automatic and human evaluation. The paper assesses the coverage of the generated results. The paper also evaluates the generation results in different settings, including longer contexts and llama2 fine-tuning. The paper also evaluates the results with GPT4.
3. The paper provides prompts, generation results, and evaluation results in the Appendix.

**Weaknesses:**

1. The plan intermediate step for generation is not new. The idea has been published in previous papers (Wang et al., 2023). Their approach incorporates arithmetic, commonsense, and symbolic reasoning. The
2. The paper fails to provide the statistics of the new dataset, MXS-2308. Including a qualitative error analysis for each error type across all proposed baselines would be beneficial. The paper also omits details on human evaluation, such as inter-annotator agreement and annotator profiles.
3. The paper fails to include any code for reproduction.


Wang, L., Xu, W., Lan, Y., Hu, Z., Lan, Y., Lee, R. K. W., & Lim, E. P. (2023). Plan-and-solve prompting: Improving zero-shot chain-of-thought reasoning by large language models. ACL 2023.

**Questions:**

Why learned plan is only implemented for GPT3.5?

---

### Official Review · Reviewer_y3EM · 2023-11-06

**Soundness:** 3 good
**Presentation:** 3 good
**Contribution:** 1 poor
**Rating:** 1
**Confidence:** 4

**Summary:**

The paper proposes an innovative plan-based prompting approach, which substantially improves the quality of literature reviews generated by LLM by decomposing the task into planning and generation phases.

**Strengths:**

In general, this paper is well organized and easy to follow.

**Weaknesses:**

(1) Problem Formulation: While the generation of literature reviews is indeed enhanced from the experiment results, the process does not encompass literature retrieval, ranking, or weighting, which are critical aspects of literature review creation.

(2) Limited Novelty and Contributions:  The idea of separating content planning and surface realization is totally not new in text generation. The technical contributions beyond prompting are not extensively detailed. The proposed data corpus, while valuable for the current study, may quickly become outdated due to the rapid evolution of LLMs.

(3) Problematic Evaluation: The automatic evaluation relies on ROUGE metrics, which may not be fully indicative of the text's factual accuracy or the presence of hallucinations.

(4) Lack of Insightful Conclusions: The conclusions drawn from the study do not extend into broader implications.

Typo:  We address this key problem by operating in the setting were LLMs are prompted to only use information from a specific set of actual papers to be cited… were → where

Typo: related works → related work

**Questions:**

(1) How to ensure that the plan prompts cover all related concepts in a hierarchical manner?

(2) Why the proposed method is unable to generate some paragraphs in the related work? Are they due to the inherent limitations of the LLM, the complexity of the source material, or the plan generation process itself?

---

### Meta-Review · Area_Chair_cH1J · 2023-12-13

**Metareview:**

The paper explores the zero-shot capabilities of large language models in generating literature reviews. It introduces a writing-plan-based intermediate task for this purpose, evaluated using both automatic and human assessments across various models. Some strengths highlighted by reviewers include:

The creation of new, useful datasets and a comprehensive investigation (ViNE).
Extensive comparisons conducted across different model types and settings (N3Vy).
Important real-world applications (CcYt).

However, the paper also faces several key concerns. These include issues with the formulation and limited novelty. Reviewer y3EM points out problems with the evaluation, such as the absence of literature retrieval, ranking, or weighting in the process, and the reliance on a potentially outdated data corpus. Other major concerns are:

Clarity issues and missing details, including dataset statistics, methodology specifics, and information on human evaluation (u18m, ViNE, CcYt).
Concerns about potential leakage of ground truth information and lack of evidence supporting claims of reduced hallucinations in generated outputs (ViNE).
Questions about the realism and novelty of the plan-based method, along with insufficient experimental details (N3Vy).
Poor presentation and a confusing experimental setup (CcYt).

**Justification For Why Not Higher Score:**

Reviewers have raised many major concerns.

**Justification For Why Not Lower Score:**

NA

---

### Decision · Program_Chairs · 2024-01-16

Reject